# Interpreting Layered Neural Networks via Hierarchical Modular Representation

## Abstract

Interpreting the prediction mechanism of complex models is currently one of the most important tasks in the machine learning field, especially with layered neural networks, which have achieved high predictive performance with various practical data sets. To reveal the global structure of a trained neural network in an interpretable way, a series of clustering methods have been proposed, which decompose the units into clusters according to the similarity of their inference roles. The main problems in these studies were that (1) we have no prior knowledge about the optimal resolution for the decomposition, or the appropriate number of clusters, and (2) there was no method with which to acquire knowledge about whether the outputs of each cluster have a positive or negative correlation with the input and output dimension values. In this paper, to solve these problems, we propose a method for obtaining a hierarchical modular representation of a layered neural network. The application of a hierarchical clustering method to a trained network reveals a tree-structured relationship among hidden layer units, based on their feature vectors defined by their correlation with the input and output dimension values.

## 1 Introduction

To construct a method for interpreting the prediction mechanism of complex statistical models is currently one of the most important tasks in the machine learning field, especially with layered neural networks (or LNNs), which have achieved high predictive performance in various practical tasks. Due to their complex hierarchical structure and the nonlinear parameters that they use to process the input data, we cannot understand the function of a trained LNN as it is, and we need some kind of approximation method to convert the original function of an LNN into a simpler interpretable representation.

Recently, various methods have been proposed for interpreting the function of an LNN, and they can be roughly classified into (1) the approximation of an LNN with an interpretable model, and (2) the investigation of the roles of the partial structures constituting an LNN (e.g. units or layers). As for approach (1), various methods have been investigated for approximating an LNN with a linear model (Lundberg & Lee, 2017; Nagamine & Mesgarani, 2017; Ribeiro et al., 2016) or a decision tree (Craven & Shavlik, 1996; Johansson & Niklasson, 2009; Krishnan et al., 1999; Thiagarajan et al., 2016). For image classification tasks in particular, methods for visualizing an LNN function have been extensively studied in terms of which part of an input image affects the prediction result (Ancona et al., 2018; Bach et al., 2015; Shrikumar et al., 2017; Simonyan et al., 2014; Smilkov et al., 2017; Springenberg et al., 2015; Sundararajan et al., 2017). Approach (2) has been studied by several authors who examined the function of a given part of an LNN (Alain & Bengio, 2017; Luo et al., 2016; Raghu et al., 2017; Zahavy et al., 2016). There has also been an approach designed to automatically extract the cluster structure of a trained LNN (Watanabe et al., 2017b;a; 2018a) based on network analysis.

Although the above studies have made it possible to provide us with an interpretable representation of an LNN function with a fixed resolution (or number of clusters), there is a problem in that we do not know in advance the optimal resolution for interpreting the original network. In the methods described in the previous studies (Watanabe et al., 2017a;b; 2018a;b;c), the unit clustering results may change greatly with the cluster size setting, and there is no criterion for determining the optimal

cluster size. Another problem is that the previous studies could only provide us with information about the magnitude of the relationship between a cluster and each input or output dimension value, and we could not determine whether this relationship was positive or negative.

In this paper, we propose a method for extracting a hierarchical modular representation from a trained LNN, which provides us with both hierarchical clustering results with every possible number of clusters and the function of each cluster. Our proposed method mainly consists of three parts: (a) training an LNN for a given data set based on error back propagation, (b) determining the feature vectors of each hidden layer unit based on its correlation with the input and output dimension values, and (c) the hierarchical clustering of the feature vectors. Unlike the clustering methods in the previous studies, the role of each cluster is computed as a centroid of the feature vectors defined by the correlations in step (b), which enables us to know the representative mapping performed by the cluster in terms of both sign and magnitude for each input or output dimension.

We show experimentally the effectiveness of our proposed method in interpreting the internal mechanism of a trained LNN, by applying it to two kinds of data sets: the MNIST data set that contains digit image data and a sequential data set of food consumer price indices. Based on the experimental results for the extracted hierarchical cluster structure and the role of each cluster, we discuss how the overall LNN function is structured as a collection of individual units.

## 2 TRAINING A LAYERED NEURAL NETWORK

An LNN can be trained to approximate the input-output relationship of an arbitrary data set $(x, y)$ that consists of input data $x \in \mathbb{R}^M$ and output data $y \in \mathbb{R}^N$, by using a function $f(x, w)$ from $x \in \mathbb{R}^M$ and a parameter $w \in \mathbb{R}^L$ to $\mathbb{R}^N$. An LNN parameter is defined by $w = \{\omega_{ij}^d, \theta_i^d\}$, where $\omega_{ij}^d$ is the connection weight between the $i$-th unit in a depth $d$ layer and the $j$-th unit in a depth $d + 1$ layer, and $\theta_i^d$ is the bias of the $i$-th unit in the depth $d$ layer. Here, $d = 1$ and $d = d_0$, respectively, correspond to the input and output layers. The LNN function $f(x, w)$ is a set of functions $\{f_j(x, w)\}$ for all output dimensions $j$, each of which is defined by $f_j(x, w) = \sigma(\sum_i \omega_{ij}^{d_0-1} o_i^{d_0-1} + \theta_j^{d_0-1})$. Here, $\sigma(x) = 1/(1 + \exp(-x))$, and $o_i^d$ is the output value of the $i$-th unit in the depth $d$ layer and $o_i^1 = x_i$ holds in the input layer. Such output values in each layer are given by $o_j^d = \sigma(\sum_i \omega_{ij}^{d-1} o_i^{d-1} + \theta_j^{d-1})$.

The purpose of training an LNN is to find an optimal parameter $w$ to approximate the true input-output relationship with a finite size training data set $\{(X_n, Y_n)\}_{n=1}^{n_1}$, where $n_1$ is the sample size. The training error $E(w)$ of an LNN is given by $E(w) = \frac{1}{n_1} \sum_{n=1}^{n_1} \|Y_n - f(X_n, w)\|^2$, where $\| \cdot \|$ is the Euclidean norm of $\mathbb{R}^N$.

Since the minimization of the training error $E(w)$ leads to overfitting to a training data set, we adopt the L1 regularization method (Ishikawa, 1990; Tibshirani, 1996) to delete redundant connection weights and obtain a sparse solution. Here, the objective function to be minimized is given by $H(w) = \frac{n_1}{2} E(w) + \lambda \sum_{d,i,j} |\omega_{ij}^d|$, where $\lambda$ is a hyperparameter used to determine the strength of regularization. The minimization of such a function $H(w)$ with the stochastic steepest descent method can be executed by an iterative update of the parameters from the output layer to the input layer, which is called error back propagation (Rumelhart et al., 1986; Werbos, 1974). The parameter update is given by

$$\Delta\omega_{ij}^{d-1} = -\eta(\delta_j^d o_i^{d-1} + \lambda \operatorname{sgn}(\omega_{ij}^{d-1})), \ \ \Delta\theta_j^d = -\eta\delta_j^d,$$

where $\delta_j^{d_0} = (o_j^{d_0} - y_j)(o_j^{d_0}(1 - o_j^{d_0}) + \epsilon_1)$, and $\delta_j^d = \sum_{k=1}^{l_{d+1}} \delta_k^{d+1} \omega_{jk}^d (o_j^d(1 - o_j^d) + \epsilon_1)$ for $d = d_0 - 1, \cdots, 2$. Here, $y_j$ is the $j$-th output dimension value of a randomly chosen $n$-th sample $(X_n, Y_n)$, $\epsilon_1$ is a hyperparameter for the LNN convergence, and $\eta$ is the step size for training time $t$ that is determined such that $\eta(t) \propto 1/t$. In the experiments, we adopt $\epsilon_1 = 0.001$ and $\eta = 0.7 \times a_1 n_1/(a_1 n_1 + 5t)$, where $a_1$ is the mean iteration number for LNN training per dataset.

# 3 HIERARCHICAL MODULAR REPRESENTATION OF LNNs

## 3.1 DETERMINING FEATURE VECTORS OF HIDDEN LAYER UNITS

To apply hierarchical clustering to a trained LNN, we define a feature vector for each hidden layer unit. Let $v_k$ be the feature vector of the $k$-th hidden layer unit in a hidden layer. Such a feature vector should reflect the role of its corresponding unit in LNN inference. Here, we propose defining such a feature vector $v_k$ of the $k$-th hidden layer unit based on its correlations between each input or output dimension. In previous studies (Watanabe et al., 2018b;c), methods have been proposed for determining the role of a unit or a unit cluster based on the square root error. However, these methods can only provide us with knowledge about the magnitude of the effect of each input dimension on a unit and the effect of a unit on each output dimension, not information about how a hidden layer unit is affected by each input dimension and how each output dimension is affected by a hidden layer unit. In other words, there is no method that can reveal whether an increase in the input dimension value has a positive or negative effect on the output value of a hidden layer unit, or whether an increase in the output value of a hidden layer unit has a positive or negative effect on the output dimension value. To obtain such sign information regarding the roles of each hidden layer unit, we use the following definition based on the correlation.

**Definition 1** (**Effect of $i$-th input dimension on $k$-th hidden layer unit**). *We define the effect of the $i$-th input dimension on the $k$-th hidden layer unit as $v_{ik}^{\text{in}}$, where*

$$v_{ik}^{\text{in}} = \frac{E\left[\left(X_i^{(n)} - E[X_i^{(n)}]\right)\left(o_k^{(n)} - E[o_k^{(n)}]\right)\right]}{\sqrt{E\left[\left(X_i^{(n)} - E[X_i^{(n)}]\right)^2\right]E\left[\left(o_k^{(n)} - E[o_k^{(n)}]\right)^2\right]}}.$$

*Here, $E[\cdot]$ represents the mean for all the data samples, $X_i^{(n)}$ is the $i$-th input dimension value of the $n$-th data sample, and $o_k^{(n)}$ is the output of the $k$-th hidden layer unit for the $n$-th input data sample.*

**Definition 2** (**Effect of $k$-th hidden layer unit on $j$-th output dimension**). *We define the effect of the $k$-th hidden layer unit on the $j$-th output dimension as $v_{kj}^{\text{out}}$, where*

$$v_{kj}^{\text{out}} = \frac{E\left[\left(o_k^{(n)} - E[o_k^{(n)}]\right)\left(y_j^{(n)} - E[y_j^{(n)}]\right)\right]}{\sqrt{E\left[\left(o_k^{(n)} - E[o_k^{(n)}]\right)^2\right]E\left[\left(y_j^{(n)} - E[y_j^{(n)}]\right)^2\right]}}.$$

*Here, $y_j^{(n)}$ is the value of the $j$-th output layer unit for the $n$-th input data sample.*

We define a feature vector of each hidden layer unit based on the above definitions.

**Definition 3** (**Feature vector of $k$-th hidden layer unit**). *We define the feature vector of the $k$-th hidden layer unit as $v_k \equiv [v_{1k}^{\text{in}}, \cdots, v_{i_0 k}^{\text{in}}, v_{k1}^{\text{out}}, \cdots, v_{kj_0}^{\text{out}}]$. Here, $i_0$ and $j_0$, respectively, represent the dimensions of the input and output data.*

**Alignment of signs of feature vectors based on cosine similarity**  The feature vectors of Definition 3 represent the roles of the hidden layer units in terms of input-output mapping. When interpreting such roles of hidden layer units, it is natural to regard the roles of any pair of units $(k_1, k_2)$ as being the same iff they satisfy $v_{k_1} = v_{k_2}$ or $v_{k_1} = -v_{k_2}$. The latter condition corresponds to the case where the $k_1$-th and $k_2$-th units have the same correlations with input and output dimensions except that their signs are the opposite, as depicted in Figure 1. To regard the roles of unit pairs that satisfy one of the above conditions as the same, we propose an algorithm for aligning the signs of the feature vectors based on cosine similarity (Algorithm 1). By randomly selecting a feature vector and aligning its sign according to the sum of the cosine similarities with all the other feature vectors, the sum of the cosine similarities of all the pairs of feature vectors increases monotonically. We show experimentally the effect of this sign alignment algorithm in Appendix 2.

## 3.2 HIERARCHICAL CLUSTERING OF UNITS IN A TRAINED LNN

Once we have obtained the feature vectors of all the hidden layer units as described in section 3.1, we can extract a hierarchical modular representation of an LNN by applying hierarchical clustering

**Algorithm 1** Alignment of signs of feature vectors based on cosine similarity

1: Let $v_k$ and $a_0$ respectively be the feature vector for the $k$-th hidden layer unit and the number of iterations.
2: **for** $a = 1$ to $a_0$ **do**
3:     Randomly choose the $k$-th hidden layer unit according to the uniform distribution.
4:     **if** $\sum_{l \neq k} \frac{v_k \cdot v_l}{\sqrt{v_k \cdot v_k}\sqrt{v_l \cdot v_l}} < 0$ **then**
5:         $v_k \leftarrow -v_k$.
6:     **end if**
7: **end for**

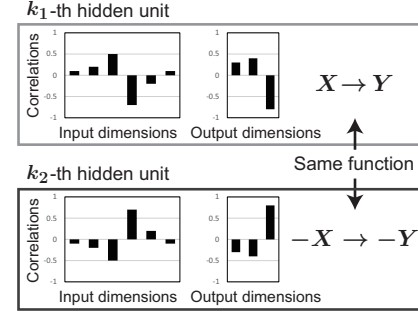

Figure 1: An example of two hidden layer units with the same function. The corresponding feature vectors are the same, except that their signs are opposite.

to the feature vectors. Among the several existing methods for such hierarchical clustering including single-link and complete-link, Ward's method (Ward, 1963) has been shown experimentally to be effective in terms of its classification sensitivity, so we employ this method in our experiments.

We start with $k_0$ individual hidden layer units, and sequentially combine clusters with the minimum *error sum of squares (ESS)*, which is given by

$$ESS \equiv \sum_m \left( \sum_{k:u_k \in C_m} \|v_k\|^2 - \frac{1}{|C_m|} \left\| \sum_{k:u_k \in C_m} v_k \right\|^2 \right), \tag{1}$$

where $u_k$ and $v_k$, respectively, are the $k$-th hidden layer unit ($k = 1, \cdots, k_0$) and its corresponding feature vector, $C_m$ is the unit set assigned to the $m$-th cluster, and $|\cdot|$ represents the cluster size. From Equation (1), the ESS is the value given by first computing the cluster size ($|C_m|$) times the variance of the feature vectors in each cluster, and then by taking the sum of all these values for all the clusters. When combining a pair of clusters ($C_{m_1}, C_{m_2}$) into one cluster, the ESS increases by

$$\Delta ESS = \frac{|C_{m_1}||C_{m_2}|}{|C_{m_1}| + |C_{m_2}|} \left\| \frac{1}{|C_{m_1}|} \sum_{k:u_k \in C_{m_1}} v_k - \frac{1}{|C_{m_2}|} \sum_{k:u_k \in C_{m_2}} v_k \right\|^2. \tag{2}$$

Therefore, in each iteration, we do not have to compute the error sum of squares for all the clusters, instead we simply have to compute the error increase $\Delta ESS$ given by Equation (2) for all the pairs of current clusters ($C_{m_1}, C_{m_2}$), find the optimal pair of clusters that achieves the minimum error increase, and combine them. We describe the whole procedure of Ward's method in Algorithm 2.

This procedure to combine a pair of clusters is repeated until all the hidden layer units are assigned to one cluster, and from the clustering result $\{C_m^{(t)}\}$ in each iteration $t = 1, \cdots, k_0 - 1$, we can obtain a hierarchical modular representation of an LNN, which connects the two extreme resolutions given by "all units are in a single cluster" and "all clusters consist of a single unit." The role of each extracted cluster can be determined from the centroid of the feature vectors of the units assigned to the cluster, which can be interpreted as a representative input-output mapping of the cluster.

## 4 EXPERIMENTS

We apply our proposed method to two kinds of data sets to show its effectiveness in interpreting the mechanism of trained LNNs. The experimental settings are detailed in the Appendix 3. In Appendix 1, we provide a qualitative comparison with the previous method (Watanabe et al., 2018c).

### 4.1 EXPERIMENT USING THE MNIST DATA SET

First, we applied our proposed method to an LNN trained with the MNIST data set (LeCun et al., 1998) to recognize 10 types of digits from input images. Before the LNN training, we sharpened the top, bottom, left and right margins and then resized the images to $14 \times 14$ pixels. Figure 2 shows sample images for each class of digits. Although our proposed method provided us with a clustering

---

**Algorithm 2** Ward's hierarchical clustering method (Ward, 1963)

---

1: Let $u_k$ and $v_k$, respectively, be the $k$-th hidden layer unit ($k = 1, \cdots, k_0$) and its corresponding feature vector, and let $\{C_m^{(t)}\}$ be the unit set assigned to the $m$-th cluster in the $t$-th iteration ($m = 1, \cdots, k_0 - t + 1$). Initially, we set $t \leftarrow 1$ and $C_m^{(1)} \leftarrow \{u_m\}$.

2: **for** $t = 2$ to $k_0 - 1$ **do**

3: $\quad (C_{m_1}^{(t-1)}, C_{m_2}^{(t-1)}) \leftarrow \arg\min_{(C_i^{(t-1)}, C_j^{(t-1)})} \Delta ESS(C_i^{(t-1)}, C_j^{(t-1)})$, where

$$\Delta ESS(C, C') \equiv \frac{|C||C'|}{|C| + |C'|} \left\| \frac{1}{|C|} \sum_{k : u_k \in C} v_k - \frac{1}{|C'|} \sum_{k : u_k \in C'} v_k \right\|^2.$$

$\quad\quad$ Here, we assume $m_1 < m_2$.

4: $\quad$ Update the clusters as follows:

$$C_m^{(t)} \leftarrow \begin{cases} C_{m_1}^{(t-1)} \cup C_{m_2}^{(t-1)} & (m = m_1) \\ C_m^{(t-1)} & (1 \leq m \leq m_2 - 1, m \neq m_1) \\ C_{m+1}^{(t-1)} & (m_2 \leq m \leq k_0 - t + 1) \end{cases}.$$

5: **end for**

---

result for all the possible resolutions or the numbers of clusters $c$, we have only plotted the results for $c = 4, 8, 16$, for ease of visibility. Figures 3 and 4, respectively, show the hierarchical cluster structure extracted from the trained LNN and the roles or representative input-output mappings of the extracted clusters. From these figures, we can gain knowledge about the LNN structure as follows.

- At the coarsest resolution, the main function of the trained LNN is decomposed into Clusters 1, 2, 3 and 4. Cluster 1 captures the input information about black pixels in the shape of a 6 and white pixels in the shape of a 7, and it has a positive and negative correlation with the output dimensions corresponding to "6" and "7", respectively. Cluster 2 correlates negatively with the region in the shape of a 9, and positively with the other areas. It has a positive correlation with the recognition of "2" and "6," and it has a negative one with "0," "4" and "9." Cluster 3 correlates positively with the black pixels in the left part of an image, and it has a positive correlation with "0," "4" and "6," and a negative correlation with "3" and "7." Cluster 4 captures the 0-shaped region, and it has a larger correlation with the output of "0" compared with the other digits.

- Cluster 2 is decomposed into three smaller clusters, 7, 8 and 9. Cluster 7 captures similar input information to Cluster 2, and it also correlates strongly with the lower area of an image. This cluster mainly affects the recognition result for "5" and "6." Cluster 8 uses the input information of the area with the shape of a 9, however, its main recognition target is "2." Cluster 9 correlates positively with the area extending from the upper right to the lower left of an image, and it correlates negatively with the digits "4" and "9."

- Cluster 8 consists of two smaller clusters, 17 and 18. Cluster 17 is mainly affected by the upper part and lower right part of an image, and the absolute value of its correlations with output dimensions are all less than 0.2, while the role of Cluster 18 is almost the same as that of Cluster 8.

### 4.2 EXPERIMENT USING THE CONSUMER PRICE INDEX DATA SET

We also applied the proposed method to an LNN trained with a data set of a consumer price index (e Stat, 2018) to predict the consumer price indices of taro, radish and carrot for a month from 36 months' input data. With this data set, we plotted the results for $c = 3, 6, 12$, where $c$ is the number of clusters. Figures 5 and 6, respectively, show the hierarchical cluster structure extracted from the trained LNN and the roles or representative input-output mappings of the extracted clusters. From these figures, we can gain knowledge about the LNN structure as follows.

- Clusters 1, 2 and 3 represent the main input-output function of the hidden layer units. Interestingly, all of these clusters have similar correlations with the output dimensions ($0 <$ radish $<$ taro $<$ carrot). However, these three clusters use different input information: Cluster 1 strongly reflects seasonal information, and its correlation is especially high with the consumer price indices of the

three vegetables one month before and one, two and three years earlier. Cluster 3 also reflects seasonal information, however, the absolute values of the correlations are less than 0.3 and it correlates strongly with the input information of eight, 20 and 32 months before. On the other hand, Cluster 2 does not use such a seasonal effect very much, and it is affected almost equally by the information of all months, except the recent information of radish from nine months before.

- Cluster 1 is composed of smaller clusters of 16 and 17. Cluster 16 is mainly used to predict the consumer price index of taro and it strongly correlates with the input information for taro from one month before and one, two and three years before. Compared with Cluster 16, Cluster 17 affects the three output dimensions more equally.

- Cluster 7 is a part of Cluster 3, and consists of smaller clusters of 11, 12 and 13. These clusters have mutually different relationships with the output dimension values: Cluster 11 correlates positively with consumer price indices of taro and carrot, and negatively with that of radish. It mainly uses recent information about carrot (within a year) and the values of taro of five, 17 and 29 months before. Cluster 13 is mainly used to predict the radish output value. It has a positive correlation with the input information for taro, radish and carrot of about six, 18 and 30 months earlier, and it has a negative correlation with values for one month before and one, two and three years before. The absolute values of the correlations between Cluster 12 and the output dimension values are less than 0.2, so, unlike with Clusters 11 and 13, it does not significantly affect the prediction result.

## 5 DISCUSSION

Here, we discuss our proposed method for obtaining a hierarchical modular representation from the perspectives of statistical evaluation and visualization.

Our proposed method provides us with a series of clustering results for an arbitrary cluster size, and the resulting structure does not change if we use the same criterion (e.g. error sum of squares for Ward's method) for evaluating the similarity of the feature vectors. However, there is no way to determine which criterion yields the optimal clustering result to represent a trained LNN, due to the fact that interpretability of acquired knowledge cannot be formulated mathematically (although there has been an attempt to quantify the interpretability for a specific task, especially image recognition (Bau et al., 2017)). This problem makes it impossible to compare different methods for interpreting LNNs quantitatively, as pointed out in the previous studies (Lipton, 2016; Doshi-Velez & Kim, 2017). Therefore, the provision of a statistical evaluation method as regards both interpretability and accuracy for the resulting cluster structure constitutes important future work.

Although we can apply our proposed method to an arbitrary network structure, as long as it contains a set of units that outputs some value for a given input data sample, the visualization of the resulting hierarchical modular representations becomes more difficult with a deeper and a larger scale network structure, since a cluster may contain units in mutually distant layers. Additionally, the number of possible cluster sizes increases with the scale (or the number of units) of a network, and so it is necessary to construct a method for automatically selecting a set of representative resolutions, instead of visualizing the entire hierarchical cluster structure.

## 6 CONCLUSION

Finding a way to unravel the function of a trained LNN is an important issue in the machine learning field. While LNNs have achieved high prediction accuracy with various data sets, their highly complex and nonlinear parameters have made it difficult to interpret their internal inference mechanism. Recent studies have enabled us to decompose a trained LNN into simpler cluster structure, however, there is no method for (1) determining the optimal number of clusters, or (2) knowing whether the outputs of each cluster have a positive or negative correlation with the input and output dimension values. In this paper, we proposed a method for extracting the hierarchical modular representation of a trained LNN, which consists of sequential clustering results with every possible number of clusters. By determining the feature vectors of the hidden layer units based on their correlations with input and output dimension values, it also enabled us to know what range of input each cluster maps to what range of output. We showed the effectiveness of our proposed method experimentally by applying it to two kinds of practical data sets and by interpreting the resulting cluster structure.

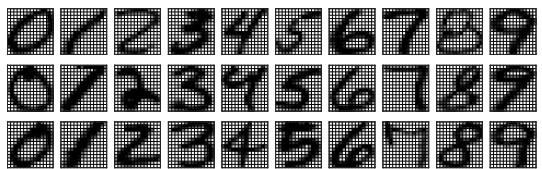

Figure 2: Input image examples of MNIST data set.

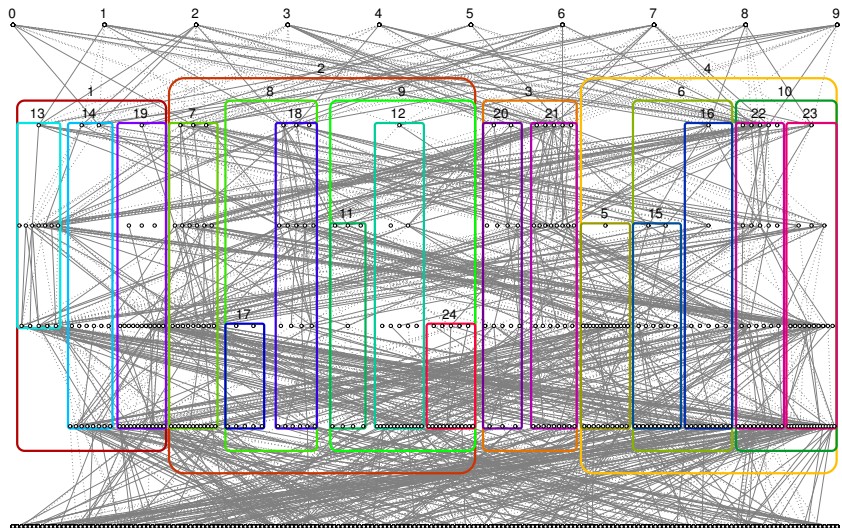

Figure 3: Hierarchical clusters of an LNN (**MNIST data set**).

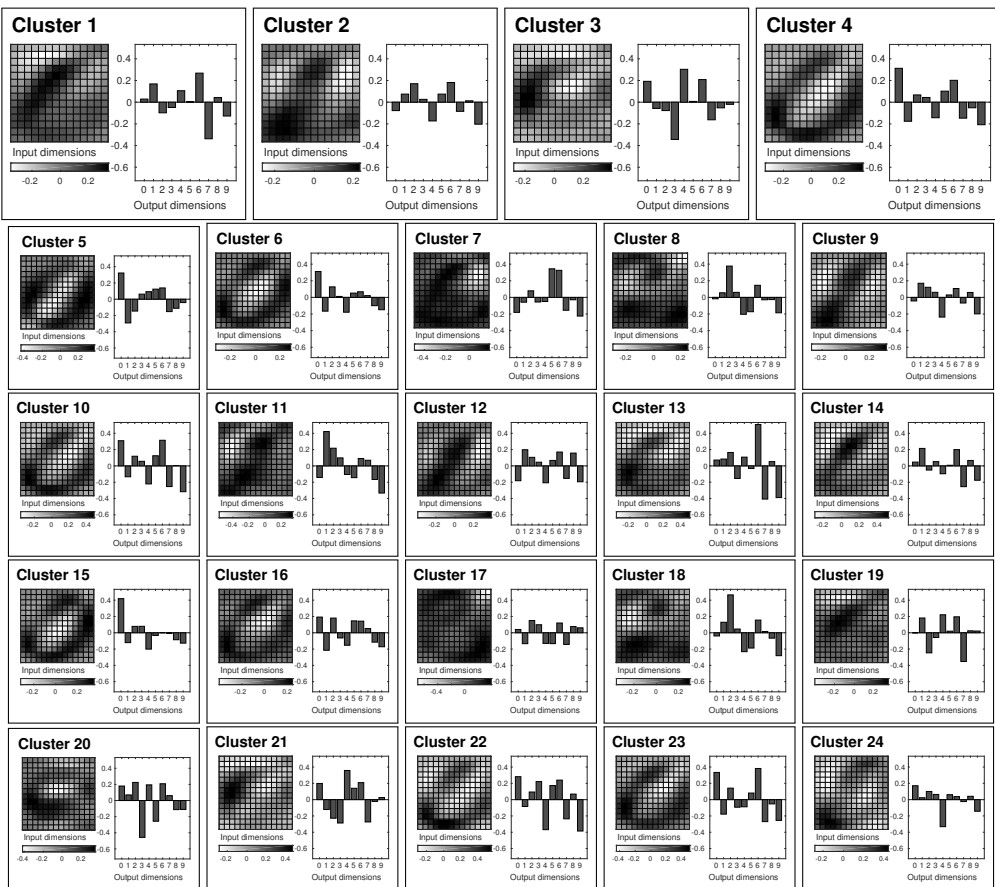

Figure 4: Representative input-output mappings of extracted clusters.

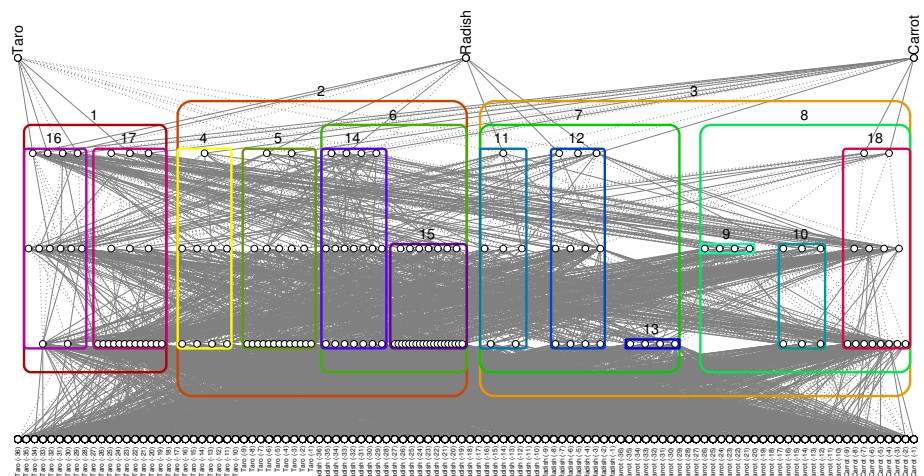

Figure 5: Hierarchical clusters of an LNN (**food consumer price index data set**).

Figure 6: Representative input-output mappings of extracted clusters.

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

## APPENDIX 1: COMPARISON WITH CLUSTERING METHOD BASED ON NON-NEGATIVE MATRIX FACTORIZATION

Here, we show the effectiveness of our proposed method by comparing it with the clustering method based on non-negative matrix factorization (or NNMF), which was proposed in a previous study (Watanabe et al., 2018c). We applied this NNMF-based clustering method to the same data sets that we used in the experiments described in section 4. In the previous study (Watanabe et al., 2018c), the feature vectors of the hidden layer units are defined by the magnitude of the effect of each input dimension value on a cluster and the effect of a cluster on each output dimension value, computed by the square root error of the unit output values. By definition, the elements of such feature vectors are all non-negative, which is a necessary condition for applying NNMF to the feature vectors.

We applied the NNMF-based clustering method to the trained network with exactly the same parameter as the network shown in Figures 3 and 5. With the MNIST data set (LeCun et al., 1998) and the data set of a consumer price index (e Stat, 2018), respectively, we decomposed the trained networks into 16 and 12 clusters. With both data sets, we set the number of iterations of the NNMF algorithm at 1000. We applied the NNMF algorithm for 10000 times, and used the best result in terms of the approximation error. Initial values of the two low-dimensional matrices were randomly chosen according to the normal distribution $\mathcal{N}(0.5, 0.5)$.

Figures 7, 8, 9, and 10 show the resulting cluster structures and the representative roles of the clusters. Comparing these figures with the results in Figures 3, 4, 5, and 6, we can observe that the previous NNMF-based method could not capture the structures of the input and output dimension values in as much detail as our proposed method, since it does not take the sign information into account. Furthermore, with the NNMF-based method, we should define the number of clusters in advance, and we cannot observe the hierarchical structure of clusters to find the optimal resolution for interpreting the roles of partial structures of an LNN.

## APPENDIX 2: EFFECT OF SIGN ALIGNMENT OF FEATURE VECTORS

Here, we discuss the effect of the sign alignment of the feature vectors based on cosine similarity (Algorithm 1).

Figures 11 shows the effect of the sign alignment of the feature vectors extracted from an LNN trained with the MNIST data set (LeCun et al., 1998). The left and center figures, respectively, show the feature vectors before and after the alignment of the signs. The right figure shows the monotonic increase of the sum of the cosine similarities through the alignment algorithm. Figure 12 shows the dendrograms of the hierarchical clustering results with the original feature vectors of Definition 3 and with the feature vectors after the alignment of the signs. From this figure, we can observe that the height of the dendrogram, which shows the similarity of all the hidden layer units, is higher with the original feature vectors than with the feature vectors after the sign alignment. In other words, it was shown that the algorithm successfully aligned the feature vectors so that they became similar to each other. Figures 13 and 14 show the effect of the sign alignment of the feature vectors extracted from an LNN trained with the food consumer price index data set (e Stat, 2018). These figures show similar results to those of the MNIST data set.

## APPENDIX 3: EXPERIMENTAL SETTINGS

Here, we detail the experimental settings. E1 and E2, respectively, represent the settings of the experiments described in sections 4.1 and 4.2.

- The training sample size $n_1$ was: 500 per class (E1), and 270 (E2).

- We normalized the input data so that the minimum and maximum values of an element, respectively, were $-1$ and 1. Similarly, we normalized the output data so that the minimum and maximum values of an element, respectively, were 0.01 and 0.99.

- The mean iteration number for LNN training per dataset $a_1$ was: 100 per class (E1), and 500 (E2).

- We generated the initial connection weights and biases of a layered neural network as follows: $\omega_{ij}^d \overset{\text{i.i.d.}}{\sim} \mathcal{N}(0, 0.5)$, $\theta_i^d \overset{\text{i.i.d.}}{\sim} \mathcal{N}(0, 0.5)$.

- The hyperparameter of the L1 regularization $\lambda$ was: $1.1 \times 10^{-5}$ (E1), and $2 \times 10^{-5}$ (E2).

- As regards the LNN training with the MNIST data set, we chose training data with the following deterministic procedure to stabilize the training. Let $Z_n^{(k)} \equiv \{X_n^{(k)}, Y_n^{(k)}\}$ be the $n$-th training data sample in class $k$. The training data were chosen in the following order:

$$Z_1^{(1)}, \cdots, Z_1^{(10)}, Z_2^{(1)}, \cdots, Z_2^{(10)}, \cdots, Z_{n_1}^{(1)}, \cdots, Z_{n_1}^{(10)},$$
$$Z_1^{(1)}, \cdots, Z_1^{(10)}, \cdots$$

- The iteration number for the alignment of the signs of the feature vectors $a_0$ was: $5000$ (E1 and E2).

- The weight removing hyperparameter $\xi$ was: $0.6$ (E1), and $0.001$ (E2) In Figures 3 and 5, we only draw connections where the absolute values of weights were $\xi$ or more.

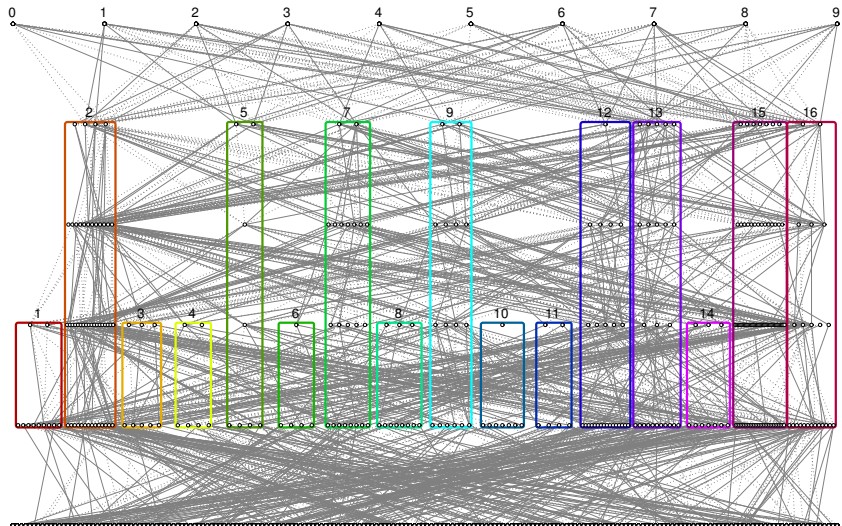

Figure 7: Cluster structure of an LNN acquired by non-negative matrix factorization (**MNIST data set**).

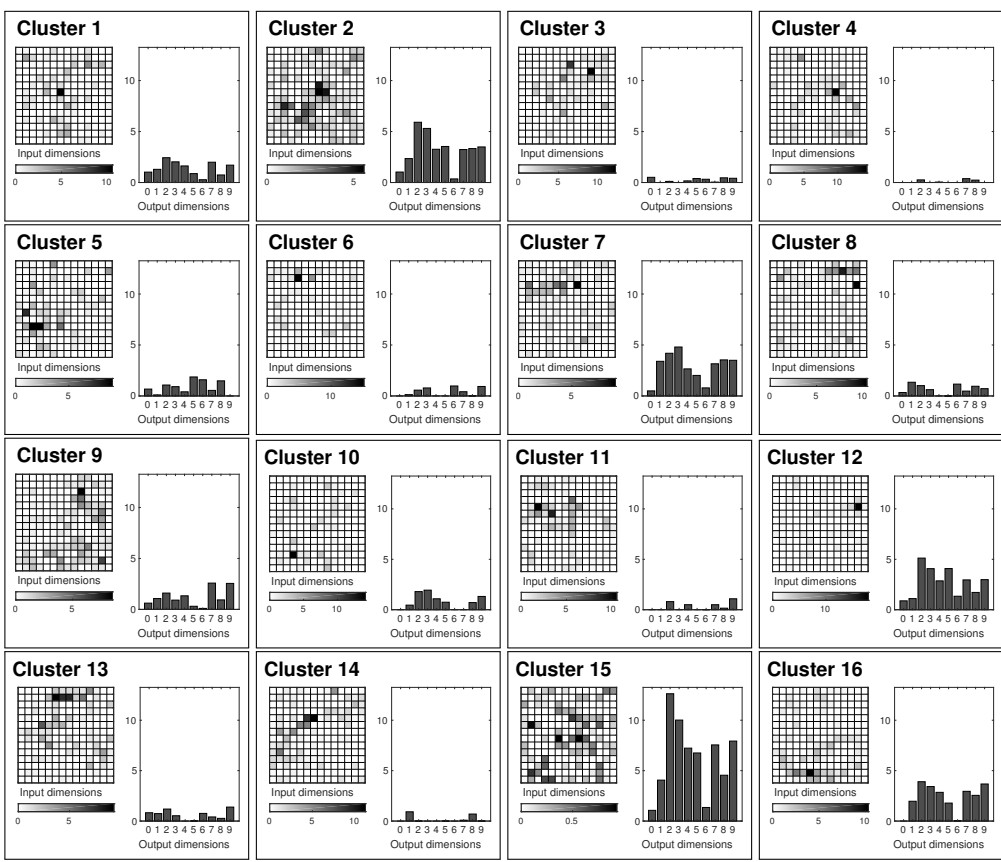

Figure 8: Representative input-output mappings of extracted clusters.

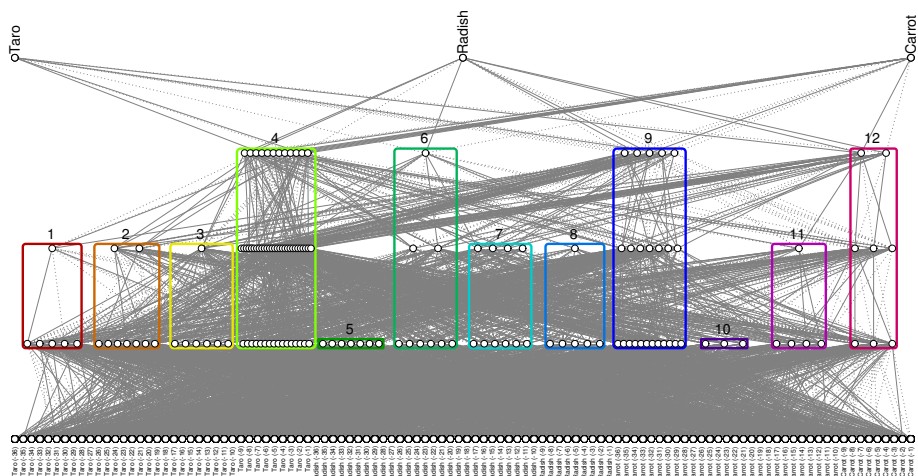

Figure 9: Cluster structure of an LNN acquired by non-negative matrix factorization (**food consumer price index data set**).

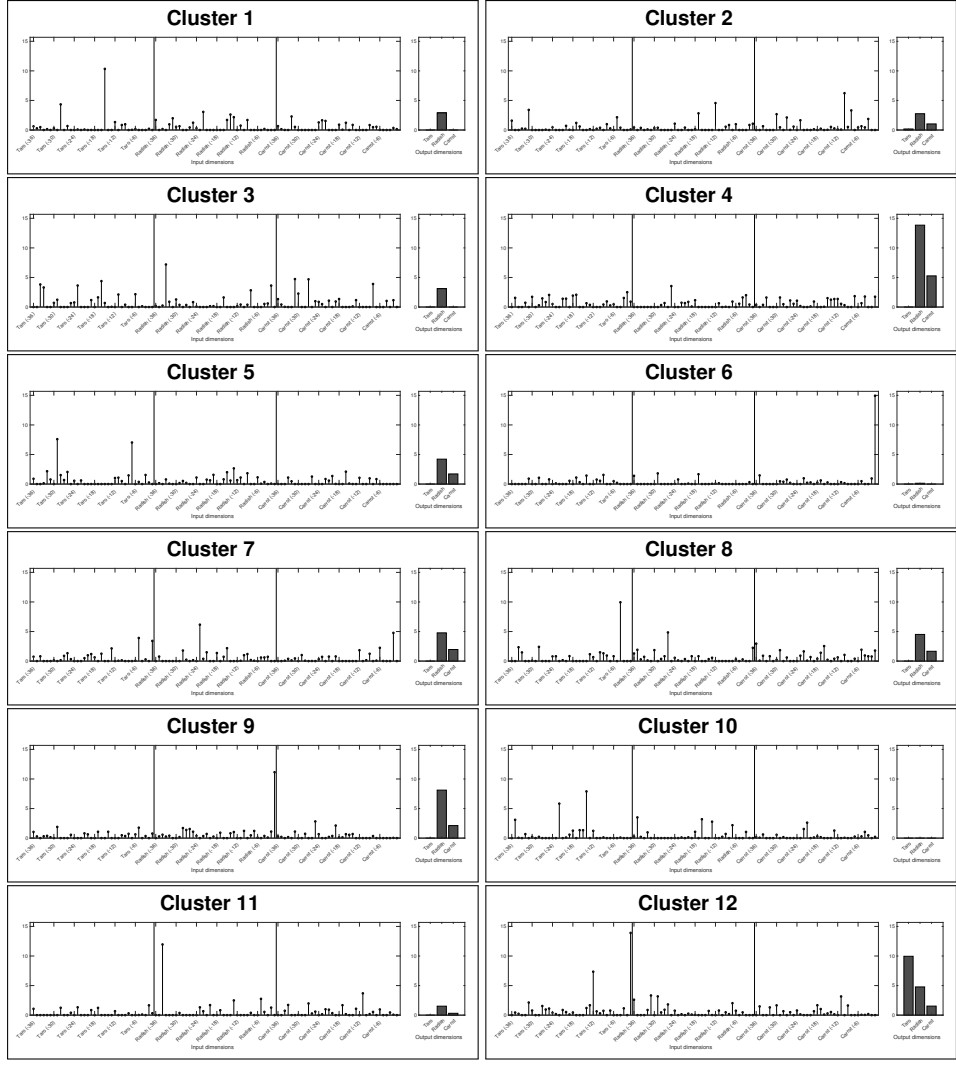

Figure 10: Representative input-output mappings of extracted clusters.

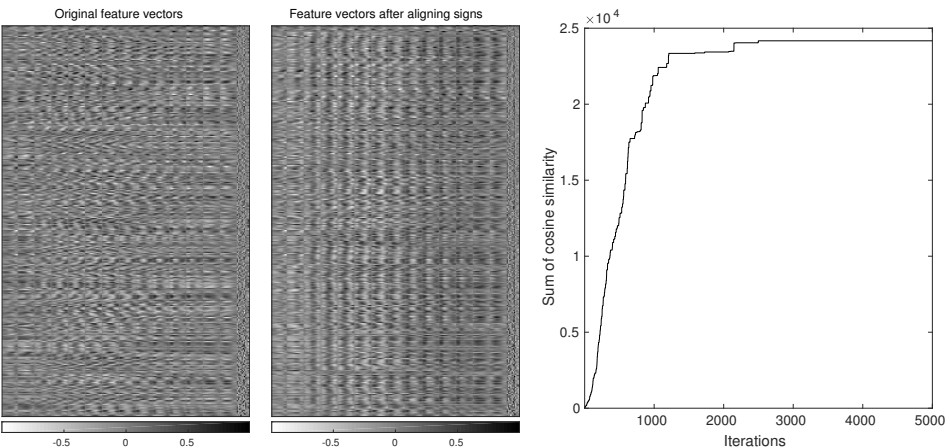

Figure 11: **Left:** Feature vectors of Definition 3. Each row corresponds to a feature vector for a hidden layer unit. **Center:** Feature vectors after the alignment of the signs. **Right:** Sum of the cosine similarities of all the pairs of feature vectors (**MNIST data set**).

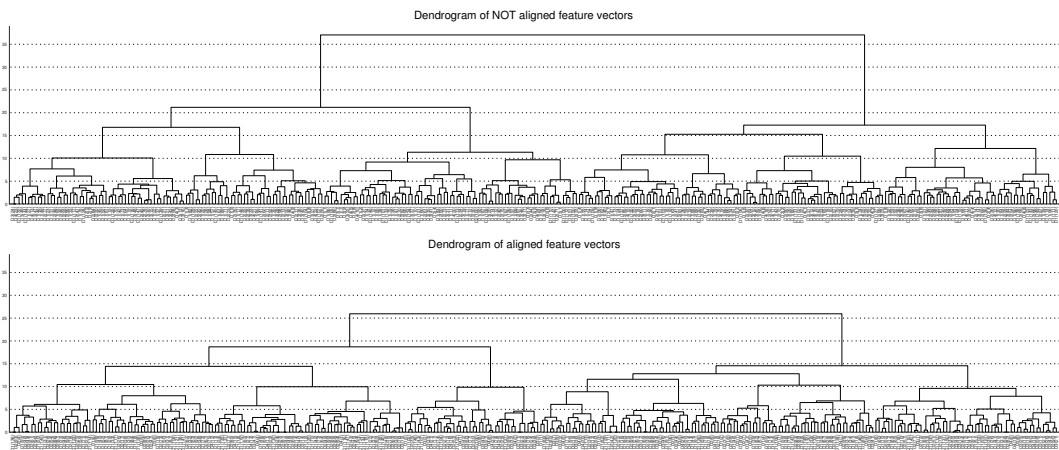

Figure 12: Dendrograms of the hierarchical clustering results with the original feature vectors of Definition 3 (top) and with the feature vectors after the alignment of the signs (bottom).

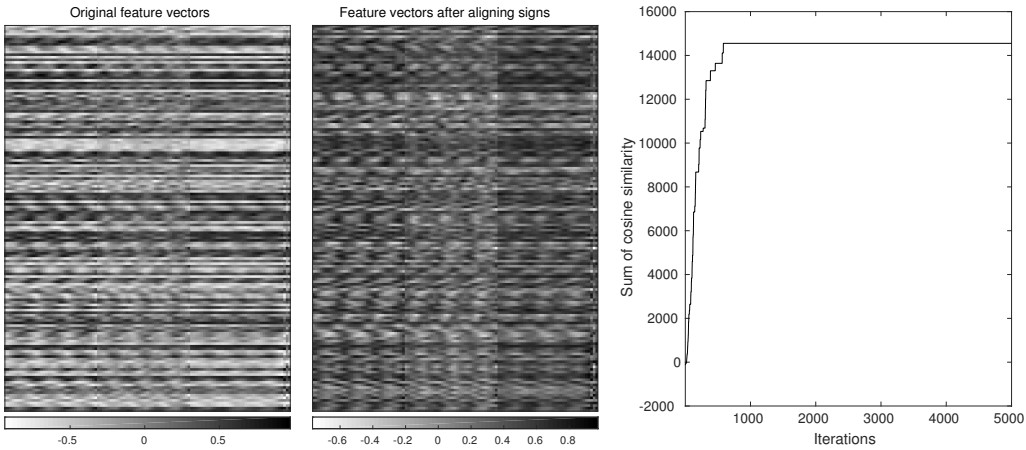

Figure 13: **Left:** Feature vectors of Definition 3. Each row corresponds to a feature vector for a hidden layer unit. **Center:** Feature vectors after the alignment of the signs. **Right:** Sum of the cosine similarities of all the pairs of feature vectors (**food consumer price index data set**).

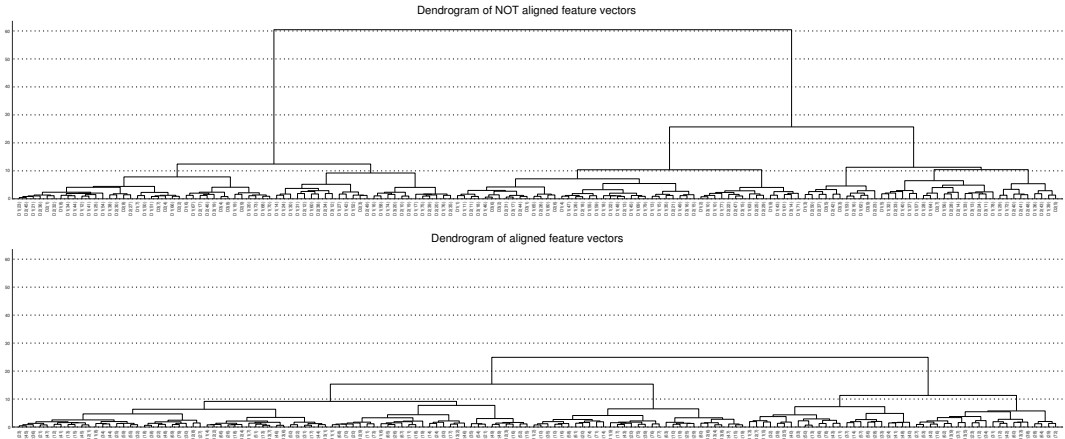

Figure 14: Dendrograms of the hierarchical clustering results with the original feature vectors of Definition 3 (top) and with the feature vectors after the alignment of the signs (bottom).

