# OpenReview forum: "Interpreting Layered Neural Networks via Hierarchical Modular Representation"
_ICLR.cc/2019/Conference_

### Official Review · AnonReviewer2 · 2018-10-30
**I recommend rejection of this paper, because 1) I do not think the proposed method achieve its purpose; 2) It is not appropriately compared with existing methods; and 3) I am not convinced that the method is designed properly.**

**Rating:** 3
**Confidence:** 4

**Review:**

In this paper, the authors try to interpret the prediction mechanism of Layered Neural Networks (LNNs). The authors proposed to first define a feature vector that represents the roles of each hidden layer unit, via computing Pearson correlation coefficient. Then a hierarchical clustering method is applied to the generated feature vectors, such that tree-structured relationships among hidden layer units are revealed.

The purpose of the paper is to understand the prediction mechanism of Layered Neural Networks (LNNs). But based on the results in the experiments, I do not think the model achieves this purpose. Given the tree structure of LNN for the MNIST data set, I am still not able to understand how this LNN distinguishes the digit 0 from other digits. I am also not able to understand why a particular sample is classified as 0 rather than 6.

In Section 1, the authors mension that there are existing clustering-based methods that interpret LNN. The authors do not compare the proposed methods with these existing methods, either quantitatively or qualitatively. So I am also not sure the contribution of this paper, provided the existing methods.

In Section 3.1, the authors state that "there is no method that can reveal whether an increase in the input dimension value has a positive or negative effect on the output value of a hidden layer unit". I do not agree with this statement, because Ross et.al (2017) has proposed to measure it via gradient, although they are trying to solve a slightly different problem. Since the output of a hidden unit is a non-linear function of the input, I am not convinced that the proposed method that computes Pearson correlation coefficient is better choise than computing the gradient.

The proposed method provides a tree structure to describe the relationships between the hidden layer units. The authors also do not illustrate why learning the tree structure is particularly important. We can also run k-means with cosine similarity on the generated vector $v$, and learn the number of clusters via Bayesian information criterion (BIC). The authors do not explain why the tree-structured clustering results are more superior than the k-means clustering results.

In summary, I recommend rejection of this paper, because 1) I do not think the proposed method achieve its purpose; 2) It is not appropriately compared with existing methods; and 3) I am not convinced that the method is designed properly.


References
Ross, Andrew Slavin, Michael C. Hughes, and Finale Doshi-Velez. "Right for the right reasons: training differentiable models by constraining their explanations." Proceedings of the 26th International Joint Conference on Artificial Intelligence (IJCAI). 2017.

---

### Official Review · AnonReviewer3 · 2018-10-31
**Unpromising approach to an important problem, understanding processing in feed-forward MLPs**

**Rating:** 3
**Confidence:** 4

**Review:**

Pros

1.	The paper is fairly clear.
2.	The problem is important: analyzing the internal computations of layered networks.
3.	The method seems to be a slight improvement on an existing method: the use of hierarchical clustering is nice.
4.	Figs. 3 and 5 superimposing the analyzed clusters on top of the network diagram are cool.

Cons

5.	The paper wastes valuable space writing out in detail the equations for backpropagation in a standard feed-forward MLP.
6.	The paper does not have an acceptable review of relevant prior work. This is particularly problematic as the proposal seems to be a rather small tweak to prior work of two 2018 papers by Watanabe et al. But there is extensive other literature attempting to address this problem, especially in the vision domain, where their main example - poor over-worked MNIST - resides.
7.	In my view, attempts to understand processing in NNs exclusively at the individual-unit level are essentially doomed at the outset. These networks crucially represent their information in distributed representations and it is joint action by multiple units rather than action by individual units that drives processing. Consider the “effect” variable analyzed in this paper, which is a simple correlation between the activity of a target hidden unit and the activity of a particular input or output unit. Suppose whenever hidden unit i is active, hidden unit j is also active, and vice versa. Now suppose j strongly drives output unit k via a connection with a large weight, while unit i has no connection at all to unit k. Then i will have a strong “effect” on k! The correlations between the activity of i and k is the same as the correlation between j and k, even though the causal interaction between i and k is nil, while the causal interaction between j and k is strong. In this especially transparent situation, it is the joint action of i and j that matters, and it so happens that this joint action has no contribution from i.
8.	So in addition to the problems arising from analyzing exclusively at the individual-unit level, there is the problem of defining “effect” by correlation instead of causation.
9.	I don’t myself gain any insight into how the MNIST network is working by looking at the clusters diagrammed in Fig. 4. There is no discussion of the fact that nearly all of their input-“effect” maps look like a slanted oval which is either on-center-off-surround or the reverse (no comment on the superficial, at least, connection to the receptive fields of neurons in the early mammalian visual system). Just how do these cluster maps explain anything?
10.	The maps for the other example, time-series of prices of root vegetables, are even more baffling, but, superficially at least, the input maps suggest the hidden units are doing Fourier analysis; even this obvious observation is not made in the paper, however.

---

### Official Review · AnonReviewer1 · 2018-11-08
**Not convinced**

**Rating:** 4
**Confidence:** 3

**Review:**

Sorry, I am not convinced by this paper.

I just don't believe that one can really gain any useful insight into neural networks by this kind of visualization.  In my opinion, all these kinds of visualization can give is the false believe that one understands what the network is doing.  (If you think about it, understanding itself is a rather vague and subjective term).  I guess my point is, these kinds of visualization don't seem to generate any actionable knowledge.  And how would one even meaningfully compare the outputs of competing methods of this general type?

---

### Meta-Review · Area_Chair1 · 2018-12-14
**Meta-Review for Hierarchical Modular Representation paper**

**Confidence:** 5
**Recommendation:** Reject

**Metareview:**

All reviewers agree to reject. While there were many positive points to this work, reviewers believed that it was not yet ready for acceptance.